Is high-intensity interval cycling feasible and more beneficial than continuous cycling for knee osteoarthritic patients? Results of a randomised control feasibility trial

http://orcid.org/0000-0001-9851-1068 Keogh Justin W. 1 2 3 jkeogh@bond.edu.au
Grigg Josephine 1
http://orcid.org/0000-0002-8447-8399 Vertullo Christopher J. 1 4
1 Faculty of Health Sciences and Medicine, Bond University , Gold Coast, QLD , Australia
2 Human Potential Centre, Auckland University of Technology , Auckland , New Zealand
3 Cluster for Health Improvement, Faculty of Science, Health, Education and Engineering, University of the Sunshine Coast , Sunshine Coast, QLD , Australia
4 Knee Research Australia , Gold Coast, QLD , Australia
Ramírez-Campillo Rodrigo
Electronic publication date: 2018 May 9
Publication date: 2018
Volume: 6
Electronic Location ID: e4738
Received 2018 Feb 24; Accepted 2018 Apr 19
Copyright: © 2018 Keogh et al.
Copyright year: 2018
Copyright holder: Keogh et al.
License: This is an open access article distributed under the terms of the Creative Commons Attribution License, which permits unrestricted use, distribution, reproduction and adaptation in any medium and for any purpose provided that it is properly attributed. For attribution, the original author(s), title, publication source (PeerJ) and either DOI or URL of the article must be cited.
License URL: https://creativecommons.org/licenses/by/4.0/

Keywords: Cycling, Feasibility, High intensity interval training, Home-based exercise, Musculoskeletal conditions

Funding: The authors received no funding for this work.

==============================
Background

Knee osteoarthritis (OA) patients often suffer joint pain and stiffness, which contributes to negative changes in body composition, strength, physical performance (function), physical activity and health-related quality of life. To reduce these symptoms and side effects of knee OA, moderate-intensity continuous training (MICT) cycling is often recommended. While resistance training is considered the optimal form of training to improve sarcopenic outcomes, it imposes higher joint loads and requires supervision, either initially or continuously by trained exercise professionals. Therefore, this pilot study sought to gain some insight into the feasibility and potential benefits of high-intensity interval training (HIIT) cycling as an alternative exercise option to MICT cycling for individuals with knee OA.

Methods

Twenty-seven middle-aged and older adults with knee OA were randomly allocated to either MICT or HIIT, with both programs involving four unsupervised home-based cycling sessions (∼25 min per session) each week for eight weeks. Feasibility was assessed by enrolment rate, withdrawal rate, exercise adherence and number of adverse effects. Efficacy was assessed by health-related quality of life (Western Ontario and McMaster Universities Osteoarthritis Index (WOMAC) and Lequesne index), physical function (Timed Up and Go (TUG), Sit to Stand (STS) and preferred gait speed) and body composition (body mass, BMI, body fat percentage and muscle mass).

Results

Twenty-seven of the interested 50 potential participants (54%) enrolled in the study, with 17 of the 27 participants completing the trial (withdrawal rate of 37%); with the primary withdrawal reasons being unrelated injuries or illness or family related issues. Of the 17 participants who completed the trial, exercise adherence was very high (HIIT 94%; MICT 88%). While only three individuals (one in the MICT and two in the HIIT group) reported adverse events, a total of 28 adverse events were reported, with 24 of these attributed to one HIIT participant. Pre–post-test analyses indicated both groups significantly improved their WOMAC scores, with the HIIT group also significantly improving in the TUG and STS. The only significant between-group difference was observed in the TUG, whereby the HIIT group improved significantly more than the MICT group. No significant changes were observed in the Lequesne index, gait speed or body composition for either group.

Discussion

An unsupervised home-based HIIT cycle program appears somewhat feasible for middle-aged and older adults with knee OA and may produce similar improvements in health-related quality of life but greater improvements in physical function than MICT. These results need to be confirmed in larger randomised controlled trials to better elucidate the potential for HIIT to improve outcomes for those with knee OA. Additional research needs to identify and modify the potential barriers affecting the initiation and adherence to home-based HIIT cycling exercise programs by individuals with knee OA.

Introduction

Osteoarthritis (OA) is a highly prevalent degenerative joint disease affecting many middle-aged and older adults, with recent global data indicating OA of the hip and knee was ranked as the 11th highest contributor to global disability and 38th highest in disability adjusted life years of the 291 health conditions analysed (Cross et al., 2014). The knee is one of the most affected osteoarthritic joints, resulting in a range of symptoms including pain and tenderness that typically limit the individuals’ physical function and mobility (Flores & Hochberg, 1988; Hootman et al., 2003). Such changes typically result in considerable physical inactivity, with a recent accelerometer cross-sectional study involving 1,111 participants reporting that only 13% of men and 8% of women with OA were meeting the recommended levels of aerobic physical activity and that an additional 40% of men and 57% of women could be classified as inactive, participating in no bouts of moderate to vigorous physical activity for more than 10 min at a time (Dunlop et al., 2011). These knee OA-related reductions in physical function and physical activity may accelerate the age-related loss of muscle mass, muscle strength and function (especially around the knee joint), referred to as sarcopenia (Cruz-Jentoft et al., 2010; Senior et al., 2015); and contribute to an increased risk of obesity and poor cardiovascular health (Roubenoff, 2000). Collectively, the symptoms and adverse events associated with the development of knee OA may further compromise health and well-being and contribute to a downward spiral into greater disability, poor health-related quality of life and further health complications (Yoshimura et al., 2012).

Several recent meta-analyses indicate that a variety of forms of exercise significantly improve pain, function and health-related quality of life in individuals with knee OA (Fransen et al., 2015; Tanaka et al., 2015; Uthman et al., 2013). While a number of different modes of exercise have been examined in the studies included in these meta- analyses, cycling and aquatic exercise are often the most commonly recommended (Arthritis Foundation) and performed (Hootman et al., 2003) exercise options for individuals with knee OA in the community. Such popularity may reflect the benefits and tolerability of these forms of exercise, with aquatic exercise and cycling imposing lower knee joint loads compared to walking, stair climbing and common resistance training exercises such as the leg extension and squat (Hall et al., 2013; Heino Brechter & Powers, 2002; Kutzner et al., 2010, 2012; Powers et al., 2014).

The benefits of cycling for middle-aged and older adults with knee OA are also well documented. These include significant improvements in health-related quality of life, as assessed by the Western Ontario and McMaster Universities Osteoarthritis Index (WOMAC) questionnaire (Alkatan et al., 2016; Salacinski et al., 2012); physical function, as assessed by gait speed (Alkatan et al., 2016; Salacinski et al., 2012) and Sit to Stand (STS) performance (Mangione et al., 1999) and body composition, as assessed by body mass, waist and hip circumference and visceral adipose tissue (Alkatan et al., 2016). As the majority of these cycling studies utilised moderate-intensity continuous training (MICT) protocols, exercise prescriptions involving higher intensity muscle contractions may better improve sarcopenic outcomes including muscle mass, strength and function (Landi et al., 2014).

We propose that the utilisation of high-intensity interval training (HIIT) may be an exercise approach that simultaneously improves sarcopenic and cardiovascular outcomes in the OA patients, while at the same time having a similar degree of feasibility and tolerability as MICT. HIIT typically requires participants to alternate short periods (∼8–60 s) of high-intensity activity with longer (∼20–90 s) recovery periods of lower intensity activity (Boutcher et al., 2013; MacInnis & Gibala, 2017; Shiraev & Barclay, 2012). Based on data obtained from other clinical populations, HIIT cycling may provide a better stimulus for overcoming the knee OA related sarcopenic outcomes (i.e. loss in leg muscle mass, strength and function (Alvarez et al., 2017; Jia et al., 2018), while still providing a range of cardiometabolic benefits than is currently achieved by MICT exercise mode such as cycling or aquatic exercise (Francois & Little, 2015; Liou et al., 2016; Ramos et al., 2015; Shiraev & Barclay, 2012).

Unfortunately, almost no research has assessed the feasibility and potential benefits of HIIT in musculoskeletal conditions like knee OA. This may reflect the concern that some health professionals may have regarding the potential musculoskeletal and cardiovascular health risks associated with HIIT. While HIIT appears well tolerated by older and middle-aged individuals with a variety of cardiovascular diseases, even when performed at home (Aamot et al., 2014; Rognmo et al., 2012), only one study appears to have assessed the safety of HIIT in arthritic populations. This recent pilot study involving young to middle-aged adults with rheumatoid arthritis (n = 7) and juvenile idiopathic arthritis (n = 11), reported no significant change in disease activity or pain during the 10 week HIIT and MICT cycling exercise program (Sandstad et al., 2015).

Therefore, this randomised control pilot trial was conducted to compare two unsupervised home-based (HIIT and MICT) cycling exercise programs in terms of their relative feasibility and ability to improve health-related quality of life, physical function and body composition in middle-aged and older knee OA patients. It was hypothesised that both forms of home-based cycling would be feasible and produce a range of benefits for the participants.

Methods

Research design

A study protocol for this trial has been previously published (Keogh, Grigg & Vertullo, 2017), with a summary of the key details provided below. According to a recent conceptual framework paper for feasibility and pilot studies (Eldridge et al., 2016), the current study can be described as a ‘randomised pilot study.’ A variety of approaches were used to recruit participants including discussions with physicians and physiotherapists as well as media stories published in selected local magazines and newsletters that had some relevance to middle-aged and older adults. As this was a randomised pilot study that focused on the feasibility of a novel exercise program that hasn’t been investigated in middle-aged and older adults with knee OA, no specific power analysis was performed.

Prior to performing the baseline assessments, all potential participants were screened for suitability to participate in the exercise program by their physician, with this supplemented by the Australian Association of Exercise and Sports Science (ESSA) pre-screening exercise form. Once the baseline assessments were completed, participants were randomised to the appropriate intervention and provided with a written explanation of how to perform their exercise program, either at home or in a gymnasium. The participants were also given a practical familiarisation on a Monark cycle ergometer on how to perform their exercise program and how to set up their bike appropriately to minimise the risk of additional knee pain. While there was no formal contact organised with the research team during the intervention period, participants were told to contact the team should any problems arise with the program, or if they had any questions.

The randomisation to either of the two cycling groups (HIIT or MICT) involved a computer-generated randomisation sequence (https://www.random.org/) performed by the lead investigator who had no interaction with the participants in relation to their assessments. This randomisation sequence was concealed in an Excel spreadsheet until it was retrieved by the research assistant immediately after completing the baseline assessment of each participant. As a result, the research assistant who informed the participants of their group allocation and conducted all the assessments was unable to be blinded to participant allocation. The participants were unable to be blinded to their participation in the exercise trial.

The protocol for this study was approved by the Bond University Research Ethics Committee (RO1776) and registered at the ANZCTR (trial registration number ACTRN12616000273482). All eligible participants provided written consent before participating in this project. The trial recruited the first participant in May 2014, with data collection completed by January 2016 as the research assistant would not be available after that time.

Participants

Males and females between the age of 40 and 80 years with a confirmed diagnosis of knee OA by an orthopaedic surgeon were eligible to participate in this study. Their physician also had to provide clearance for them to participate in the study and to state that the patient was unlikely to change their osteoarthritic management during the study. As the cycling programs were home-based, all participants also needed to have access to a stationary bicycle to be eligible to participate. Participants were allowed other comorbidities, if these comorbidities did not contraindicate home-based, unsupervised exercise. The conditions which were contraindications to participation in this study included unstable cardiac conditions, including a history of arrhythmia and cardiac ischaemia.

Exercise program

All participants were requested to perform four stationary cycling sessions per week for eight weeks, with each session ∼25 min long. The MICT group commenced each session with a 3 min warm-up at a light intensity and finished with a 2 min cooldown at a similar intensity to the warm-up. The MICT component required the participants to select a workload that they could cycle at a cadence of 60–80 rpm for 20 min at a moderate intensity, which was defined as ‘an intensity in which you are able to speak in complete sentences during the exercise. If you find yourself getting too puffed or out of breath—slow down a little.’ For HIIT participants, their training sessions commenced with a 7 min warm-up of progressively increasing intensity and concluded with a 6–7 min cool-down of light to moderate cycling. The HIIT component of the training session involved five series of high- and low-intensity cycling. For the five, 45 s high-intensity bouts, participants were requested to cycle at a higher cadence (up to 110 rpm) for 45 s using a resistance similar or slightly higher than the low-intensity recovery bouts, which were to be performed at ∼70 rpm for a duration of 90 s. The intensity of the high-intensity bouts was defined as ‘an intensity at which you felt it was quite difficult to complete sentences during the exercise.’

To minimise the potential for any adverse events relating to the initiation of the exercise program, the participants were encouraged to be somewhat conservative with respect to the intensity of their initial sessions by decreasing the recommended cadence or workload described in the previous paragraph. If they tolerated their first few exercise sessions with no cardiovascular or joint pain, they were requested to progressively increase exercise intensity to that described in the previous paragraph.

Data collection

Outcome measures were collected at baseline and at the end of the eight week cycling programs at a university clinic that the participants were required to visit. To gain some insight into the relative feasibility and safety of the HIIT compared to the MICT cycling program, participants were given a training and adverse events diary. The participants were requested to record the date of each training session they performed across each of the 8 training weeks. Similarly, the adverse events diary required the participants to state the number of any adverse effects they experienced (if any) and to provide some description of what happened.

Feasibility outcomes

Consistent with recent feasibility studies of under-researched exercise programs for a range of older clinical populations (Bossers et al., 2014; Cheema et al., 2015; Fien et al., 2016), the feasibility and safety of the two cycling protocols was quantified by the enrolment rate, withdrawal rate, adherence rate and number of adverse events (Fien et al., 2016). Enrolment rate was expressed as a percentage and calculated by dividing the number of individuals who consented to participate in the study by the total number of individuals who expressed interest in participating. The withdrawal rate (also expressed as a percentage) was calculated by dividing the number of participants who failed to complete post-testing by the number of participants who completed baseline testing. Adherence rate was calculated by dividing the number of training sessions completed by each participant by the requested number of training sessions (n = 32), with this presented as a percentage. Adverse events were defined as ‘an exercise-induced change that worsens an aspect of your condition that is greater than expected day-to-day variation,’ a definition very similar to that used previously in the study of 1,687 men and women undergoing exercise programs (Bouchard et al., 2012).

Efficacy outcomes

Efficacy outcomes included two OA-specific health-related quality of life questionnaires (WOMAC and the Lequesne Index), three objective physical performance tests (30 s STS, Timed Up and Go (TUG) and habitual gait speed tests) and a body composition assessment (via the Tanita MC-980MA body composition analyser; Tanita, Tokyo, Japan). A brief description of these efficacy outcomes is provided below.

Western Ontario and McMaster Universities Osteoarthritis Index: The WOMAC is a valid and reliable health-related quality of life questionnaire consisting of 24 items that assesses the OA patient’s degree of pain (five items), stiffness (two items) and physical function/disability (17 items) typically felt when performing a range of activities of daily living (Bellamy, 1989; Bellamy et al., 1988). The maximum score for the WOMAC was 96, with a score of 39 or greater indicative of severe arthritis (Hawker et al., 2000).

Lequesne index of severity for osteoarthritis of the knee: The Lequesne index is another valid and reliable knee OA health-related quality of life questionnaire that consists of 11 items that assesses the patient’s pain/discomfort (five items), maximum walking distance with or without walking aids (two items) and physical function/disability (four items) (Lequesne, 1991, 1997). The maximum possible score is 24, with the severity of the knee OA-related disability described as small (1–4), intermediate (5–7), serious (8–10), very serious (11–13) and extremely serious (≥14) (Lequesne, 1991, 1997).

Sit to Stand: The STS test is a valid and reliable measure of lower extremity strength and function in older populations including those with knee OA (Lord et al., 2002). Participants were asked to sit on a straight-backed, armless chair that was ∼43 cm in height and keep their arms crossed across the chest. On the word ‘Go,’ participants completed as meeting STSs as possible in 30 s. Participants performed one practice trial of ∼3–4 repetitions at a submaximal intensity prior to performing the one trial of this exercise.

Timed Up and Go: The TUG is a valid and reliable measure of functional mobility for a range of older adult populations, including those with knee OA (Podsiadlo & Richardson, 1991). Participants were required to stand up from a ∼43 cm high, armless chair and then proceed to walk around a cone 3 m away before sitting back on the chair (Podsiadlo & Richardson, 1991), with the following instructions ‘Stand-up and walk around the flagpole and sit back down on the chair at a pace comfortable for you.’ Participants were allowed one practice trial, with the best time from three timed trials used for analysis.

Habitual gait speed: Habitual gait speed was quantified using the GaitMat II pressure mat system (Model is GaitMat II; Manufacturer is EQInc, Chalfont, USA) (Rosano et al., 2008; Trehan et al., 2015). All gait speed trials were initiated 2 m (6.56 ft.) before the GaitMat II platform (3.66 m long) and finished 2 m after the GaitMat II to reduce the potential acceleration and deceleration effects on mean gait speed (Kressig & Beauchet, 2006). Participants were provided with the following instructions ‘Walk towards the end of the room at a pace that is comfortable for you’ (Fien et al., 2016). The average gait speed from three attempts was used for data analysis.

Body composition: Body composition (proportion of muscle, fat and bone) was assessed using the Tanita MC-980MA body composition analyser (Tanita, Tokyo, Japan) (Ragini et al., 2015). As the bio-electrical impedance assessment (BIA) method is sensitive to alterations in hydration, all participants were requested to be normally hydrated and to have not eaten or exercised for a period of 2 h before the BIA assessment. All participants stood in a stationary position in bare feet on the Tanita MC-980MA platform while holding the handles for a period of 30 s. According to the manufacturer’s user manual, the sensitivity of this device was 0.1 kg for total body mass, muscle mass, fat-free mass and fat mass.

Statistical analysis

Centrality and dispersion of the continuous data was reported as means and standard deviations, whereas categorical measures were reported as number and percentage. To test for whether there were significant differences at baseline between the two groups, two tailed independent t-tests were performed for continuous variables and chi-square analysis for categorical variables. The chi-square analysis was performed using an online calculator http://www.socscistatistics.com/tests/chisquare2/Default2.aspx. Two tailed paired t-tests were used to determine within-group changes, i.e. pre–post-test changes for each of the groups independently. The magnitude of the pre–post-test changes for each group were quantified using Cohen’s d effect size and 95% confidence intervals (CI) utilising an online effect size calculator with 95% CI https://effect-size-calculator.herokuapp.com/. Effect sizes were described as being small (d = 0.20–0.49), moderate (d = 0.50–0.79) and large (d ≥ 0.80). Potential between-group differences between the HIIT and MICT groups were analysed by using a two-tailed independent t-test with unequal variance on the pre–post-test change scores for each individual and outcome measure. Except where indicated otherwise, all statistical tests were performed in Microsoft Excel 2010, with statistical significance set at p ≤ 0.05.

Results

The demographic characteristics of the participants who completed the HIIT and MCIT cycle programs are described in Table 1. There were no significant differences between the groups for any of the outcome measures at baseline (p = 0.09–0.94).

Table 1 Baseline characteristics of the participants who completed the trial.

	HIIT (n = 9)	MICT (n = 8)	Between group p-value	All (n = 17)	
Age (years)	59.1 (6.7)	66.1 (8.8)	0.09	62.4 (8.3)	
Gender (M/F)	3/6	1/7	0.31	4/13	
Height (cm)	170.0 (6.2)	165.5 (6.3)	0.16	167.9 (6.5)	
Body mass (kg)	78.5 (13.5)	77.8 (23.0)	0.94	78.2 (18.0)	
BMI	27.0 (4.0)	28.2 (6.9)	0.69	27.6 (5.4)	
Osteoarthritis side (Both/L/R)	4/3/2	4/0/4	0.67	8/3/6	
Duration of diagnosis (years)	4.6 (5.8)	4.9 (3.2)	0.91	4.7 (4.6)	
Prior Surgery	5	6	0.40	11	
WOMAC	36.1 (15.0)	34.8 (15.5)	0.85	35.5 (14.8)	
Lequesne index	8.8 (4.3)	9.6 (3.9)	0.69	9.0 (4.2)	
Notes:

All results are mean (standard deviation) except bold.

HIIT, high-intensity interval training cycling; MICT, moderate-intensity continuous training cycling; M, male; F, female; L, left; R, right.

A summary of the key feasibility outcomes are presented in Fig. 1 and Table 2. As seen in Fig. 1, a total of 27 individuals from the 50 (54% enrolment rate) who were invited/expressed interest to participate were enrolled in this study. As only 17 of the 27 participants completed the trial, the trial had a withdrawal rate of ∼37%, with the most common reasons provided for withdrawal reflecting unrelated illness, unrelated injury or family related issues.

Figure 1 Participant CONSORT flow diagram.

Table 2 Exercise adherence and adverse events.

	HIIT (n = 9)	MICT (n = 8)	All (n = 17)	
Adherence (%)	94 (8)	88 (12)	91 (10)	
Adverse events (number)	26	2	28	
Participants reporting adverse events (number)	2	1	3	
Notes:

All results are mean (standard deviation) except where indicated.

HIIT, high-intensity interval training cycling; MICT, moderate-intensity continuous training cycling.

As summarised in Table 2, a high level of exercise adherence was observed for the HIIT (94%) and MICT (88%) groups. Three participants (one in the MICT group and two in the HIIT group) reported adverse events during the eight week study, with the one individual in the MICT group reporting two adverse events relating to discomfort and pain they felt was caused by the bicycle seat. In contrast, 26 adverse events were reported for the HIIT group. Of these 26 adverse events, 24 were reported by one individual who stated that the cycling program aggravated their Bakers cyst behind their knee. Interestingly, this individual still completed 25 of the requested 32 exercise sessions.

The potential efficacy of the two home-based cycling programs with respect to improving the participants’ health-related quality of life, physical performance (function) and body composition is summarised in Table 3. The HIIT group demonstrated significant pre–post-test improvements in WOMAC, TUG and STS performance over the course of the training program; with the MICT demonstrating a significant pre–post-test improvement in the WOMAC only. Based on effect size calculations, all of these significant improvements could be described as moderate effects, with the exception of the WOMAC change for the HIIT group which would be considered a large effect. When comparing the changes between the two groups, the only significant between-group difference was observed for the TUG, whereby the HIIT group improved to a significantly greater extent than the MICT group. No significant improvements were observed in the Lequesne index or any of the body composition outcomes for either of the two groups.

Table 3 Training-related changes in health-related quality of life, functional performance and body composition.

	HIIT (n = 9)	MICT (n = 8)		
Pre-test	Post-test	Pre–post HIIT p-value	Pre–post HIIT effect size (95% CI)	Pre-test	Post-test	Pre–post MICT p-value	Pre–post MICT effect size (95% CI)	Between-group p-value	
WOMAC	36.1 (15.0)	21.2 (14.6)*	0.005	0.91 (0.28–1.70)	34.8 (15.5)	22.9 (14.4)*	0.006	0.71 (0.21–1.36)	0.829	
Lequesne index	8.8 (4.3)	6.7 (3.9)	0.102	0.46 (−0.10 to 1.10)	9.6 (3.9)	8.4 (4.6)	0.081	0.25 (−0.04 to 0.59)	0.707	
TUG (s)	8.9 (2.0)	7.8 (1.1)*†	0.004	0.62 (0.14–1.20)	9.1 (1.9)	9.7 (3.5)	0.401	−0.19 (−0.69 to 0.28)	0.043	
Sit to Stand (reps)	11.1 (2.2)	13.1 (2.7)*	0.012	0.73 (0.17–1.43)	9.4 (1.9)	10.6 (3.5)	0.095	0.38 (−0.10 to 0.92)	0.417	
Gait speed (m/s)	1.21 (0.18)	1.21 (0.14)	0.883	0.00 (−0.54 to 0.54)	1.11 (0.17)	1.14 (0.20)	0.706	0.14 (−0.68 to 0.99)	0.685	
Body mass (kg)	78.5 (13.5)	78.7 (13.6)	0.605	−0.01 (−0.02 to 0.00)	77.8 (23.0)	77.8 (22.1)	0.978	0.00 (−0.03 to 0.03)	0.715	
BMI (kg/m2)	27.0 (4.0)	26.6 (3.9)	0.426	0.09 (0.05–0.15)	28.2 (7.3)	28.2 (6.9)	0.810	0.00 (−0.04 to 0.04)	0.784	
Body fat (%)	30.9 (5.6)	31.4 (5.7)	0.197	−0.11 (−0.93 to 0.69)	36.0 (9.0)	36.6 (9.1)	0.099	−0.05 (−0.17 to 0.04)	0.967	
Muscle mass (kg)	51.6 (10.7)	51.3 (10.5)	0.179	−0.02 (−0.07 to 0.12)	45.9 (7.4)	45.5 (7.2)	0.136	−0.05 (−0.16 to 0.05)	0.861	
Notes:

All results are mean (standard deviation).

HIIT, high-intensity interval training cycling; MICT, moderate-intensity continuous training cycling.

* Significant pre–post-test (within-group) effect.

† Significant between-group effect favouring the HIIT group.

Discussion

The primary aim of this pilot study was to examine and compare the feasibility and safety of unsupervised, home-based HIIT and MICT cycling in middle-aged and older adults with knee OA; with the secondary aim being to gain some insight into the relative efficacy of these forms of cycling for improving health-related quality of life, physical performance and body composition.

The results of the current study are heterogeneous with regards to the feasibility of the two home-based cycling programs for middle-aged and older individuals with knee OA. For example, the research team took 1.5 years to identify the 50 potential middle-aged and older individuals with knee OA. The relatively slow recruitment to the study may reflect a variety of factors, perhaps including concerns from potential participants, medical practitioners and physiotherapists that the HIIT exercise program may aggravate knee OA symptoms or perhaps that relatively few individuals with knee OA had access to a suitable stationary bicycle. Qualitative and/or quantitative studies examining these potential barriers, as well as the facilitators and motives to home-based HIIT cycling may need to be conducted as has been done recently for individuals with knee OA performing group-based aquatic exercise programs (Fisken et al., 2014, 2015), to further improve these recruitment rates.

It must also be acknowledged that only 54% of the potentially interested 50 individuals enrolled in this exercise study. While this enrolment rate is relatively low it should be noted that a total of 15 potential participants (30% of those were initially interested) were deemed ineligible, with the majority of these individuals being located too far away to come in for baseline and post-tests or due to a lack of access to a stationary bike. Furthermore, the 54% enrolment rate of the current study was slightly higher than the 44% and 46%, respectively reported by Mangione et al. (1999) and Rewald et al. (2015), who also performed cycling interventions for middle-aged and older adults with knee OA. Such relatively low enrolment rates further support the need for more research needs to investigate the barriers to enrolment in such cycling studies.

Our results also indicated that only 17 of the 27 (63%) of the original participants completed the cycling program, with most of the reasons for withdrawal not related to the exercise program. This withdrawal rate was similar to Salacinski et al. (2012) who reported 68% of their original participants completed the project; but substantially lower than other cycling studies that observed 83–95% completion rates (Alkatan et al., 2016; Mangione et al., 1999; Rewald et al., 2015). For those participants who continued in our two cycling programs, a very high level of exercise adherence was reported for both the MICT (88%) and HIIT (92%) groups. Such values appear to be consistent (Alkatan et al., 2016; Mangione et al., 1999; Salacinski et al., 2012) or greater (Rewald et al., 2015) than reported in previous cycling studies involving individuals with knee OA.

Analysis of the adverse event data indicated that only three participants (one in the MICT and two in the HIIT group) reported any adverse events, defined as ‘an exercise induced change that worsens an aspect of your condition that is greater than expected day-to-day variation’ over the course of the intervention. Two of these individuals each reported two adverse events, with one individual in the HIIT group unfortunately reporting 24 adverse events over the course of the 25 exercise sessions they performed. While the proportion of participants reporting adverse events (HIIT, 2 out of 9, 22%; MICT, 1 out of 8, 13%) and a total of 24 adverse events for just one participant in the HIIT group appears very high, it needs to be remembered that this is an unsupervised exercise trial involving middle-aged and older adults with knee OA. Further, it could be argued that the severity of these adverse events was quite low as the even the individual who reported 24 further adverse events continued to perform the majority of their requested exercise sessions even though such cycling aggravated their Bakers cyst. While such adverse events may therefore not truly reflect the individual’s knee OA diagnosis, some caution may need to be made when prescribing cycling, especially of a HIIT nature to individuals with Bakers cysts; and to perhaps include Bakers cysts as an exclusion criteria for future cycling studies.

Comparison of our adverse data to that of literature was also difficult as some cycling studies did not report such data (Alkatan et al., 2016; Mangione et al., 1999) and even those studies that did, no clear definition of adverse events was often given (Rewald et al., 2015; Salacinski et al., 2012). It was also observed that some studies stated that no adverse events occurred but then indicated that a number of individuals withdrew from the study due to knee pain (Alkatan et al., 2016; Salacinski et al., 2012), although it was not explicitly stated whether such knee pain was considered an adverse event or whether it was related to the exercise program.

According to a recent review by Wellsandt & Golightly (2018), adherence and the level of improvement resulting from an exercise program for individuals with knee OA may be influenced by a variety of factors including the participant’s preference for the degree of supervision and mode of exercise. This suggests that the true feasibility of the home-based HIIT and MICT cycling observed in the present study could be higher in individuals with knee OA who are able to select their preferred exercise activity. Such a finding would suggest that clinicians and exercise professionals should endeavour to find out the exercise preferences of their patients and match these where possible to the efficacy evidence reported in randomised controlled trials (RCTs) and meta-analyses.

Regarding the efficacy outcomes, both cycling programs demonstrated some significant benefits for the participants. Both groups significantly improved their health-related quality of life as measured by the WOMAC, but not the Lequesne index. The significant improvements in their WOMAC score (HIIT: 14.9 points; MICT: 11.9 points) were comparable or greater than the 10.6–11.6 point improvements reported in the literature for other cycling exercise studies involving knee OA participants (Alkatan et al., 2016; Salacinski et al., 2012). The significant increases in the WOMAC score in the current study appear clinically significant as they exceed the nine point improvement identified by Tubach et al. (2005) as being a clinically relevant change. While the improvements in the Lequesne index were non-significant (HIIT: p = 0.102; MICT: p = 0.081), both groups reported small effect size improvements the HIIT group had a reduction of ∼2 points on the scale that resulted in their overall group being classified as ‘intermediate disability’ at post-test compared to their baseline classification of ‘serious disability’ (Lequesne, 1991, 1997).

The tendencies for the participants to report improved health-related quality of life, including reduced pain, stiffness and disability were consistent with some of the functional changes reported for the HIIT group. Specifically, the HIIT group significantly improved their TUG and STS performance, with the change in TUG performance significantly greater than the MICT group. The magnitude of these changes for the HIIT group (TUG: −1.1 s; STS: +2 repetitions) would appear clinically significant based on the minimally detectable change of 1.1 s for the TUG (Alghadir, Anwer & Brismée, 2015) and 1.64 repetitions for the STS (Gill & McBurney, 2008) reported in previous studies for individuals with knee and/or hip OA. It must however be acknowledged that neither exercise group significantly improved their habitual gait speed over the 3.66 m course. Such a lack of change in gait speed was consistent with the low (40% of heart rate reserve) and high (70% of heart rate reserve) intensity MICT cycle training groups in the study of Mangione et al. (1999), but inconsistent with two cycling studies for individuals with knee OA who reported increases of between 0.08 and 0.20 m/s over distances of 3.66–6 m (Alkatan et al., 2016; Salacinski et al., 2012).

The lack of any significant changes in body composition for either of the two groups in the current study appears relatively inconsistent with the very limited number of cycling studies assessing these outcomes for individuals with knee OA. Specifically, Alkatan et al. (2016) reported that the cycling group significantly improved body mass (−1.5 kg), waist circumference (−3 cm), hip circumference (−2 cm) and visceral adipose tissue (−0.1 kg); although no significant changes were observed for BMI, body fat percentage or lean tissue mass. The discrepancy between the results of the current study and Alkatan et al. (2016) may reflect two primary factors. The first is that since Alkatan et al. (2016) used dual-energy X-ray absorptiometry (DEXA) compared to our study’s BIA, there is the potential that the increased sensitivity of the DEXA was required to observe such body composition changes over relatively small training periods. Further, the duration of the cycling program for Alkatan et al. (2016) was 1.5 times longer (12 vs 8 weeks) than the current study. The potential for cycling, particularly the novel HIIT cycling program examined in this study to improve body composition may also be influenced by the exercise preferences of the individuals with knee OA. In a recent review, Wellsandt & Golightly (2018) found that exercise preference (e.g. degree of supervision and exercise mode) contributes to exercise adherence and to the magnitude of change in body composition outcomes with exercise in individuals with OA. Specifically, studies which randomly allocated participants to an exercise program reported no significant change in body composition, whereas the one study that allowed participants to choose their exercise program (Loew et al., 2017), reported significant reductions in body mass; with those adhering to the walking program also reporting significantly greater reductions in waist circumference.

There were however several limitations associated with the current study. The first limitation reflects the self-report nature of much of the data including the WOMAC and Lequesne index as well as the training and adverse events diary. Even though the participants typically completed their training and adverse event diaries, the actual exercise dose performed is always difficult to quantify during home-based exercise, as no measure of external (power output) or internal (heart rate) workload was collected from the participants. It is also acknowledged that the home-based nature of the intervention may not have optimised the potential benefits of the HIIT program as a trained exercise professional supervising the exercise sessions would typically monitor and progressively increase workloads over the course of the training program. The sample size was also relatively small, which may affect our ability to detect within and between group significant differences and the potential generalisability of the trial findings. Nevertheless, the sample size in the study was greater (Rewald et al., 2015) than or somewhat comparable (Mangione et al., 1999; Salacinski et al., 2012) to some other cycling studies involving middle-aged and older adults with knee OA.

Conclusion

Our results are somewhat supportive of the feasibility of home-based HIIT and MICT cycling programs for middle-aged and older adults with knee OA, with the enrolment rate and adherence rate for both the HIIT and MICT groups comparable to other cycling studies involving similar populations (Alkatan et al., 2016; Rewald et al., 2015; Salacinski et al., 2012). The number of adverse events (HIIT: two of nine participants, 26 adverse events; MICT: one of eight participants, two adverse events) appeared very high, although 24 of these adverse events were reported by one HIIT individual with a Bakers cyst on their knee, who still completed 25 of the requested 32 exercise sessions. Efficacy data demonstrated significant benefits in health-related quality of life (WOMAC) for both groups, with the HIIT group also reporting significant increases in functional performance as assessed by TUG and STS. The major limitation of this study is the small sample size, which may limit our statistical power and increase our chance of type II error. This suggests that larger scale RCTs should further investigate the feasibility and efficacy of HIIT compared to MICT cycling for a variety of knee OA sub-populations; including studies comparing standard supervised and home-based exercise programs.

Supplemental Information

Supplemental Information 1 Descriptive and inferential statistics.

Click here for additional data file.

Supplemental Information 2 Consort checklist.

Click here for additional data file.

Supplemental Information 3 Study protocol paper.

Click here for additional data file.

Additional Information and Declarations

Competing Interests

Author Contributions

Human Ethics

Clinical Trial Ethics

Clinical Trial Registration

Data Availability

Justin W. Keogh is an Academic Editor for PeerJ.

Justin W. Keogh conceived and designed the experiments, analysed the data, contributed reagents/materials/analysis tools, prepared figures and/or tables, authored or reviewed drafts of the paper, approved the final draft.

Josephine Grigg performed the experiments, analysed the data, contributed reagents/materials/analysis tools, prepared figures and/or tables, authored or reviewed drafts of the paper, approved the final draft.

Christopher J. Vertullo conceived and designed the experiments, contributed reagents/materials/analysis tools, authored or reviewed drafts of the paper, approved the final draft.

The following information was supplied relating to ethical approvals (i.e. approving body and any reference numbers):

The Bond University Research Ethics Committee granted ethical approval to carry out the study within its facilities (ethical application RO1776).

The following information was supplied relating to ethical approvals (i.e. approving body and any reference numbers):

The Bond University Research Ethics Committee granted ethical approval to carry out the study within its facilities (ethical application RO1776).

The following information was supplied regarding Clinical Trial registration:

ACTRN12616000273482.

The following information was supplied regarding data availability:

The raw data are provided in the Supplemental Files.

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
