# Peer review of "Is high-intensity interval cycling feasible and more beneficial than continuous cycling for knee osteoarthritic patients? Results of a randomised control feasibility trial"

_PeerJ, doi:10.7717/peerj.4738_

## Round 0.1 · original submission · Major Revisions

Dear authors:

Your manuscript was evaluated by two expert reviewers. Please address the reviewers' concerns in detail, and note that the comments of Reviewer 2 are in an attached PDF

Reviewer 1 ·

Basic reporting

- The article has been written in English clearly and accurately. No major errors are observed.
- The introduction shows sufficient background to understand the possible functional benefits of a high intensity interval training (HIIT) and a therapeutic alternative for older adults with knee osteoarthritis (OA).
- I suggest this modification to the title: Is high-intensity interval cycling feasible and more beneficial than continuous cycling for knee osteoarthritic patients? - Results of a randomised control feasibility trial.
- Table 1: Apply t-test for independent samples in the baseline characteristics of the groups.

Experimental design

- The manuscript complies with the Aims and Scope of PeerJ.
- The research question is well defined.
- The research complies with the ethical code of the international research community and includes a record of the protocol: “The protocol for this study was approved by the Bond University Research Ethics Committee (RO1776) and registered at the ANZCTR (trial registration number ACTRN12616000273482)”. Two groups were included: an exercise group (EG) and a control group (CG).
- The research has criteria that may increase the likelihood of type 1 error: the HIIT group presents less age and a higher proportion of male participants than the MICT group. This difference in age and sex can generate better results in favor of the experimental group (HIIT), as can be seen in the results of Table 3 (TUG). It is suggested to apply a t-test for independent samples to determine if there are significant differences between the groups in the baseline characteristics.
- L237: The abbreviation BIA is observed for the first time, without indicating the complete word.

Validity of the findings

- The results are novel and significant for the understanding and effectiveness of the training / rehabilitation protocols of patients with knee OA. However, the results and conclusions should be taken with caution, since the research has a high probability of committing a type 1 error. Consider the comment presented in "Experimental design".
- L272-278: Two of nine participants experienced adverse effects on the HIIT group, which corresponds to 18% of the participants analyzed. This corresponds a considerable percentage and even more when the sample size is small. I recommend being cautious when supporting the HIIT group over the MICT in relation to the reduction of adverse effects, considering that the main purpose of the investigation was to examine and compare the feasibility and safety. I recommend including a sentence on this comment between L328-338.

Additional comments

The research is novel and original since it attempts to solve a recurrent research problem in patients with knee OA. This research tries to understand the possible functional benefits of a HIIT and a therapeutic alternative for older adults with knee osteoarthritis (OA) through variables associated with the reliability and functionality of the patient. However, the research has criteria that may increase the likelihood of type 1 error: the HIIT group presents less age and a higher proportion of male participants than the MICT group. That can generate a high probability of observing significant results when they really are not.

·

Basic reporting

Please to read my general and specific comments attached in the document.

Experimental design

Please to read my general and specific comments attached in the document.

Validity of the findings

Please to read my general and specific comments attached in the document.

Additional comments

Please to read my general and specific comments attached in the document.

---

## Round 0.2 · accepted · Accept

No further comments. Congratulations to you and your team.

·

Basic reporting

I agree with the final version submitted by the authors.

Experimental design

I agree with the final version submitted by the authors.

Validity of the findings

I agree with the final version submitted by the authors.

Additional comments

I agree with the final version submitted by the authors.